# Neuroinflammation Is Associated with GFAP and sTREM2 Levels in Multiple Sclerosis

**DOI:** 10.3390/biom12020222

**Published:** 2022-01-27

**Authors:** Federica Azzolini, Luana Gilio, Luigi Pavone, Ennio Iezzi, Ettore Dolcetti, Antonio Bruno, Fabio Buttari, Alessandra Musella, Georgia Mandolesi, Livia Guadalupi, Roberto Furlan, Annamaria Finardi, Teresa Micillo, Fortunata Carbone, Giuseppe Matarese, Diego Centonze, Mario Stampanoni Bassi

**Affiliations:** 1IRCCS Neuromed, 86077 Pozzilli, Italy; federica.azzolini@gmail.com (F.A.); gilio.luana@gmail.com (L.G.); gi19gi82@gmail.com (L.P.); ennio.iezzi@neuromed.it (E.I.); ettoredolcetti@hotmail.it (E.D.); antonio.bruno91@yahoo.it (A.B.); fabio.buttari@gmail.com (F.B.); m.stampanonibassi@gmail.com (M.S.B.); 2Synaptic Immunopathology Lab, IRCCS San Raffaele, 00163 Roma, Italy; alessandra.musella@uniroma5.it (A.M.); georgia.mandolesi@uniroma5.it (G.M.); livia.guadalupi@gmail.com (L.G.); 3Department of Human Sciences and Quality of Life Promotion, University of Rome San Raffaele, 00166 Rome, Italy; 4Clinical Neuroimmunology Unit, Institute of Experimental Neurology (INSpe), Division of Neuroscience, San Raffaele Scientific Institute, 20132 Milan, Italy; furlan.roberto@hsr.it (R.F.); finardi.annamaria@hsr.it (A.F.); 5Neuroimmunology Unit, IRCCS Fondazione Santa Lucia, 00179 Rome, Italy; teresa.micillo2@unina.it (T.M.); f.carbone@ieos.cnr.it (F.C.); 6Laboratory of Immunology, Institute of Endocrinology and Experimental Oncology, National Research Council, 80131 Naples, Italy; giuseppe.matarese@unina.it; 7Treg Cell Lab, Department of Molecular Medicine and Medical Biotechnologies, University of Naples Federico II, 80131 Naples, Italy; 8Department of Systems Medicine, Tor Vergata University, 00133 Rome, Italy

**Keywords:** multiple sclerosis (MS), neuroinflammation, neurodegeneration, microglia, astroglia, soluble triggering receptor expressed on myeloid cells-2 (sTREM-2), glial fibrillary acid protein (GFAP), cytokines, cerebrospinal fluid (CSF) biomarkers

## Abstract

**Background**: Astrocytes and microglia play an important role in the inflammatory process of multiple sclerosis (MS). We investigated the associations between the cerebrospinal fluid (CSF) levels of glial fibrillary acid protein (GFAP) and soluble triggering receptors expressed on myeloid cells-2 (sTREM-2), inflammatory molecules, and clinical characteristics in a group of patients with relapsing-remitting MS (RRMS). **Methods**: Fifty-one RRMS patients participated in the study. Clinical evaluation and CSF collection were performed at the time of diagnosis. The CSF levels of GFAP, sTREM-2, and of a large set of inflammatory and anti-inflammatory molecules were determined. MRI structural measures (cortical thickness, T2 lesion load, cerebellar volume) were examined. **Results**: The CSF levels of GFAP and sTREM-2 showed significant correlations with inflammatory cytokines IL-8, G-CSF, and IL-5. Both GFAP and sTREM-2 CSF levels positively correlated with age at diagnosis. GFAP was also higher in male MS patients, and was associated with an increased risk of MS progression, as evidenced by higher BREMS at the onset. Finally, a negative association was found between GFAP CSF levels and cerebellar volume in RRMS at diagnosis. **Conclusions**: GFAP and sTREM-2 represent suitable biomarkers of central inflammation in MS. Our results suggest that enhanced CSF expression of GFAP may characterize patients with a higher risk of progression.

## 1. Introduction

Multiple sclerosis (MS) is a chronic autoimmune disease of the central nervous system (CNS) characterized by neuroinflammation and neurodegeneration often coexisting from the earliest stages of the disease. The pathogenesis of relapsing-remitting (RR) MS is characterized by infiltration of autoreactive lymphocytes, activation of resident immune cells, and release of a number of inflammatory mediators leading to axonal damage and neurodegeneration [1].

Microglia and astrocytes physiologically play important homeostatic functions such as supporting neuronal activity, development, and survival [2]. However, they may undergo functional changes upon inflammation or brain damage; indeed, distinct astrocytic and microglial reactive phenotypes with variable neuroprotective or proinflammatory polarization have been recognized [3,4]. Proinflammatory microglia show enhanced phagocytic activity and increased production of proinflammatory cytokines [5], and activated astrocytes participate in scar formation, secrete inflammatory mediators, and have also been linked to synaptic degeneration and glutamate dysregulation [4,6,7]. In MS, activated microglia and astrocytes may enhance neurodegenerative pathways, leading to disability progression [8]. Accordingly, glial fibrillary acid protein (GFAP) and the soluble triggering receptor expressed on myeloid cells 2 (sTREM-2), two established biomarkers of astroglial and microglial activation, have been proposed as prognostic tools in MS [9,10,11,12]. Both GFAP and sTREM-2 CSF and serum levels have been correlated with clinical disability and neuroradiological activity in MS [8,12,13,14]; moreover, serum levels of GFAP have been associated with a progressive course, higher EDSS, older age, longer disease duration [15]. Increased CSF inflammation promotes disease activity and increased prospective disability in RRMS patients [16,17]. Moreover, proinflammatory chemokines and cytokines may drive pathological activation of microglia and astrocytes, leading to progression in MS. Here we investigated in a group of treatment naive RRMS patients whether the CSF levels of GFAP and sTREM-2 are associated with central inflammation, quantified by measuring the intrathecal levels of specific proinflammatory and anti-inflammatory cytokines at the time of diagnosis. Moreover, we explored possible correlations between the CSF levels of GFAP and sTREM-2, clinical characteristics, and structural MRI measures.

## 2. Materials and Methods

### 2.1. MS Patients

A group of 51 RRMS patients admitted to the Neurology Clinic of Neuromed Hospital and diagnosed as RRMS was enrolled. The study was approved by the Ethics Committee of IRCCS Neuromed (protocol 06/2017). All patients gave written informed consent to the study.

The diagnosis of RRMS was based on published criteria [18]. All patients were newly diagnosed and clinical evaluation, brain and spine MRI and CSF withdrawal were performed within 24 h during the diagnostic work-up. Demographic and clinical data were also collected. Disease duration was estimated as the number of months from the first episode of focal neurological dysfunction indicative of MS to the time of diagnosis. Clinical disability was evaluated using the using Expanded Disability Status Scale (EDSS) [19]. BREMS (Bayesian Risk Estimate for Multiple Sclerosis) at onset was calculated [20].

### 2.2. CSF Collection and Analysis

CSF was collected by lumbar puncture, centrifuged at 1300 rpm for 10 min, and stored at −80 °C. At the time of CSF collection, all patients were treatment-naïve and none of them had previously been treated with steroids. All CSF samples were tested for oligoclonal bands (OCB) detection.

CSF GFAP and sTREM-2 levels were measured by using Human ProcartaPlex multiplex immunoassay kits (Thermo Fisher, Santa Clara, California), according to the manufacturer’s instructions. Fluorescence intensity was measured with the Luminex 200 analyzer (Luminex Corporation, Austin, TX, USA) and data were analyzed using the xPonent 3.1 software (Luminex Corporation, Austin, TX, USA).

Analysis of CSF proinflammatory and anti-inflammatory cytokines and chemokines was performed using a Bio-Plex multiplex cytokine assay (Bio-Rad Laboratories, Hercules, CA, USA), according to the manufacturer’s instructions. Samples were analyzed in triplicate. Concentrations were calculated according to a standard curve generated for the specific target and expressed in picograms per millilitre. The following CSF molecules were examined: interleukin (IL)-1β, IL-2, IL-4, IL-5, IL-6, IL-7, IL-8, IL-9, IL-10, IL-12, IL-13, IL-15, IL-1 receptor antagonist (ra), G-CSF, GM-CSF, IP-10, TNF, MIP1a, MCP1, IFN-γ.

### 2.3. MRI

All patients underwent a standard acquisition protocol with 1.5 or 3 Tesla brain MRI scan with gadolinium (Gd) administration (0.2 mL/Kg IV) with the following images: dual-echo proton density, fluid-attenuated inversion recovery (FLAIR), T2-weighted spin-echo images, and pre-contrast and post-contrast T1-weighted spin-echo. A neuroradiologist, blinded to clinical data, evaluated the presence of Gd enhancing lesions.

For the analysis of brain structural measures, in 22 MS patients, standard brain MRI acquisition protocol by using a 3T MR scanner (GE Signa HDxt, GE Healthcare, Milwaukee, WI, USA), including the following MR series: a 3D Spoiled Gradient Recalled (SPGR) T1-weighted sequence (178 contiguous sagittal slices, voxel size 1 mm × 1 mm × 1 mm, TR 7 ms, Inversion Time 450 ms) and a 3D FLAIR sequence (208 contiguous sagittal slices, voxel size 0.8 mm × 0.8 mm × 0.8 mm, TR 6000 ms, Inversion Time 1827 ms). We first segmented white matter lesions from FLAIR and T1 images by using the lesion growth algorithm as implemented in version 2.0.15 of the lesion segmentation tool (www.statistical-modelling.de/lst.html) for SPM12 (https://www.fl.ion.ucl.ac.uk/spm). Furthermore, we used the computational anatomy toolbox (CAT12, version 916, https://dbm.neuro.uni-jena.de/cat/) to extract individual cortical thickness values from lesion-filled MR images. Finally, we used the volBrain pipeline [21] to perform cerebellar volume calculation and lesionBrain pipeline [22] to estimate the lesion load for each patient.

### 2.4. Statistical Analysis

The Kolmogorov–Smirnov test was used to assess the normality distribution of data. Categorical variables were presented as number (n) and frequency (%). Continuous data were shown as median (interquartile range, IQR). To test possible associations between nonparametric variables, Spearman’s non-parametric correlation was used. Partial Spearman’s correlations were used to test the correlation between two variables, while controlling for the effect of other confounding factors (i.e., age, sex). The χ^2^ test was used to explore the association between categorical variables. Differences in continuous variables among two groups were evaluated by nonparametric Mann–Whitney U test. A *p*-value ≤ 0.05 was considered statistically significant. When testing correlations between multiple CSF molecules, Benjamini–Hochberg (B-H) procedure was used to decrease the false discovery rate and avoid Type I errors (false positives). Analyses were performed using IBM SPSS Statistics for Windows (IBM Corp., Armonk, NY, USA).

## 3. Results

### 3.1. Correlation between CSF Inflammatory Molecules and GFAP/sTREM2 Levels

The clinical characteristics of MS patients are shown in Table 1.

Infiltrating immune cells release proinflammatory cytokines in MS brains, that could, in turn, activate astroglia and microglial cells. Thus, we explored the correlations between the CSF levels of GFAP and sTREM2 and inflammatory cytokines.

Correlations between CSF inflammatory molecules and GFAP and sTREM2 are shown in the Appendix A. Significant correlations were found between the CSF levels of GFAP and IL-5, IL-8, G-CSF, and IFNγ (Figure 1A). These correlations were significant also controlling for the effect of sex and age at LP: IL-5 (partial Spearman’s r = 0.297, *p* = 0.038), IL-8 (partial Spearman’s r = 0.496, *p* < 0.001), G-CSF (partial Spearman’s r = 0.543, *p* < 0.001), and IFNγ (partial Spearman’s r = 0.385, *p* = 0.006).

Significant correlations were found between the CSF levels of sTREM-2 and IL-5, IL-8, G-CSF, IL-13, and IL-9 (Figure 1B). These correlations were significant also controlling for the effect of sex and age at LP: IL-5 (partial Spearman’s r = 0.287, *p* = 0.045), IL-8 (partial Spearman’s r = 0.430, *p* = 0.002), G-CSF (partial Spearman’s r = 0.548, *p* < 0.001), IL-9 (partial Spearman’s r = 0.452, *p* = 0.001), and IL-13 (partial Spearman’s r = 0.352, *p* = 0.011).

Finally, according to the idea that astroglia and microglial cells are activated in parallel and interact mutually, we also found a strong correlation between GFAP and sTREM-2 CSF levels (Figure 1C).

### 3.2. Association between Clinical Characteristics of MS Patients and the CSF Levels of GFAP and sTREM2

Significant correlations were found between age at LP and the CSF levels of both GFAP (Spearman’s r = 0.431, *p* = 0.002, N = 51) and sTREM-2 (Spearman’s r = 0.372, *p* = 0.007, N = 51) (Figure 2A,B). Furthermore, higher levels of GFAP were found in male patients [F, median (IQR) = 53.69 (40.25–84.52); M, median (IQR) = 86.23 (56.1–113.26); *p* = 0.007]. Conversely, no significant differences emerged between male and female patients in CSF levels of sTREM-2 (*p* = 0.424) (Figure 2C,D).

No significant associations emerged between the CSF levels of GFAP and sTREM-2 and disease duration (*p* = 0.304 and *p* = 0.748 respectively), radiological activity at LP (*p* = 0.789 and *p* = 0.878 respectively), and the presence of OCB (*p* = 0.184 and *p* = 0.102 respectively). In addition, no significant correlations were observed between the CSF levels of GFAP and sTREM-2 and EDSS at LP (*p* = 0.431 and *p* = 0.903 respectively).

A significant correlation was only found between GFAP CSF levels and BREMS at onset (Spearman’s r = 0.501, *p* < 0.001, N = 47). Conversely, no significant correlations emerged between BREMS at onset and CSF levels of sTREM-2 (*p* = 0.142) (Figure 2E,F).

### 3.3. Correlation between MRI Measures and GFAP/sTREM-2 CSF Levels

In 22 RRMS patients the associations between CSF levels of GFAP and sTREM-2 with structural MRI measures were explored. A significant correlation emerged between GFAP and both cortical thickness (Spearman’s r = −0.449, *p* = 0.036), and cerebellar volume (Spearman’s r = −0.427, *p* = 0.047). Conversely no significant association emerged between GFAP and T2 lesion load (*p* = 0.414) (Figure 3A). In addition, no significant correlations were found between CSF sTREM-2 concentrations and MRI measures (Figure 3B), although a non-significant negative trend was observed with the cortical thickness (Spearman’s r = −0.391, *p* = 0.072).

The correlation between GFAP and cerebellar volume was significant, also controlling for the effect of age at diagnosis and sex (partial Spearman’s r = −0.450, *p* = 0.046). Conversely, the correlation between GFAP and cortical thickness was not significant when controlling for the effect of age at diagnosis and sex (partial Spearman’s r = −0.353, *p* = 0.127).

## 4. Discussion

Microglia and astrocytes play an important role in the pathogenesis of MS and have received considerable attention for their possible involvement in progressive disease phenotypes [23,24]. MS is characterized by a transition from a homeostatic-anti-inflammatory to a pro-inflammatory phenotype in both the microglial and astroglial compartments [25,26]. A proinflammatory CSF milieu could possibly promote and sustain microglial and astroglial activation, predisposing to neurodegeneration and disease progression. We explored in a group of RRMS patients at diagnosis, the association between the CSF levels of GFAP and sTREM-2, which are known biomarkers of astroglial and microglial activation and the concentrations of a large set of proinflammatory and anti-inflammatory molecules.

GFAP is a cytoskeletal protein expressed by astrocytes and represents an established marker of astroglial activation and astrogliosis [27,28,29]. TREM-2 is a surface receptor expressed by microglial cells [30]. Activation of TREM-2 mediates anti-inflammatory responses by modulating cytokine release, proliferation, migration, and phagocytosis of apoptotic cells and myelin debris [9]. Soluble TREM2 (sTREM2), detectable in the CSF and serving as a surrogate measure of microglial activity [31], has been associated with inhibition of the anti-inflammatory function of the surface receptor [9].

Consistent with the hypothesis that both microglia and astrocytes participate in the induction and maintenance of the inflammatory response in MS, we found a strong positive correlation between CSF levels of GFAP and sTREM-2. Moreover, we found a positive correlation between GFAP and sTREM-2 and the levels of several inflammatory cytokines, suggesting an association between microglial and astroglial activation and CSF inflammation.

GFAP and sTREM-2 have been investigated in MS as useful tools to monitor disease progression [9,10,11,12]. Previous studies have shown higher levels of GFAP and TREM-2 in the CSF of patients with RR and progressive MS compared with patients with non-inflammatory neurological disorders [8,9,13,32,33,34]. Serum and CSF GFAP concentrations have also been associated with clinical disability [8,13,14] and with radiological activity [12]. Moreover, in patients with progressive MS, serum GFAP concentrations have been associated with age and EDSS, and with neurofilament light (NFL) levels [10]. These data highlight that GFAP and sTREM-2 may represent useful prognostic biomarkers in MS.

In our study, the positive correlations emerged between the CSF concentrations of GFAP and sTREM-2 and specific inflammatory molecules suggest a link between CSF inflammatory levels and both astroglial and microglial activation in MS.

Previous studies conducted in RRMS patients showed that CSF inflammation may be associated with a worse disease course. Elevated expression of proinflammatory molecules, including IL-1β, IL-2, IL-6, and IL-8 have been associated with higher prospective disease activity, disability, and neurodegeneration in MS [35,36,37].

We have found that among all cytokines examined, IL-5, IL-8, and G-CSF have shown correlation with both biomarkers, whereas the other cytokines correlated with only one of the two biomarkers. These data do not allow us to clearly establish whether there are specific associations between either of the two explored markers and specific cytokines. Moreover, among the cytokines that showed positive correlations, there are some important proinflammatory cytokines, but also molecules with immunomodulatory action and others whose role in MS is still unclear. IL-8, G-CSF, and IFNγ have been previously associated with MS pathogenesis and have shown several proinflammatory interactions also with microglia and astrocytes. IL-8 is a major proinflammatory molecule involved in several inflammatory and non-inflammatory neurological conditions [38]. Preclinical studies have demonstrated that IL-8 is released by microglia and astrocytes in response to different inflammatory stimuli [39,40]. This cytokine strongly promotes the chemotaxis and activation of immune cells into the CNS [41]. In MS, increased CSF levels of this molecule have been associated with greater prospective disease activity and disability [17,37]. G-CSF is primarily produced by T cells, macrophages, and endothelial cells, and can also be secreted in the CNS by astrocytes, microglia, and neurons [40,42]. This molecule binds to this receptor (G-CSFR) expressed by astrocytes, microglia, and neurons [42,43], inducing microglial proliferation and activation [44,45]. Elevated levels of G-CSF have been found in the CSF of patients with neuromyelitis optica compared with patients with other neurological non-inflammatory conditions [46], and it has also been reported that treatment with G-CSF may promote clinical worsening in patients with MS [47]. IFNγ has also been implicated in MS and EAE pathogenesis, as increased expression of this proinflammatory molecule has been reported in EAE [48] and in MS patients [49]. IFNγ plays a pivotal role in regulating the immune system, also through the activation of receptors expressed on glia and neurons [50]. Microglia respond to IFNγ with increased expression of surface molecules, greater production of proinflammatory molecules, and enhanced phagocytic and antigen-presenting activity [51,52].

We have also found associations between immunomodulatory molecules and GFAP and sTREM-2 CSF concentrations. IL-5 is mainly produced by Th2 lymphocytes, is functionally related to IL-4 and IL-13, and is involved in humoral immune responses. In MS patients, IL-5 and IL-13 are induced after treatment with glatiramer acetate, suggesting a protective role of these molecules [53]. IL-13 has been associated with neuroprotective effects in MS, however, the role of IL-5 in MS is still unclear [54,55]. Finally, we found a significant positive correlation between IL-9 and TREM-2 CSF levels. IL-9 is released mostly by Th9 cells and in a smaller proportion by Th2, Th17, Treg, and mast cells. IL-9 binds to its receptor (IL-9R), which is strongly expressed by macrophages and microglial cells [56,57,58]. Previous studies have shown enhanced serum levels of IL-9R in autoimmune conditions (i.e., lupus erythematosus) [59]. In EAE and MS, IL-9 has been associated with anti-inflammatory activities and neuroprotection [60,61]. IL-9 reduced the activation of macrophages and microglia, inhibited the release of proinflammatory molecules, and promoted an anti-inflammatory phenotype [58].

The correlation between GFAP and sTREM-2 and levels of various inflammatory cytokines is consistent with a crosstalk between CSF inflammation and activation of microglia and astroglia in MS. The correlation shown with proinflammatory and anti-inflammatory molecules suggests that, especially in the early stages of the disease, proinflammatory and immunoregulatory processes coexist. The association between GFAP and sTREM2 and cytokine levels could therefore signal an immune activation in the CNS.

To explore possible associations between microglial and astroglial activation and clinical features of MS, we investigated whether the CSF levels of GFAP and sTREM2 correlated with specific demographic and clinical characteristics at the time of MS diagnosis. An association between age at MS diagnosis and both GFAP and sTREM-2 CSF levels was found. In addition, we evidenced increased levels of CSF GFAP in male patients compared with females. These data are reflected in a positive correlation between GFAP levels and BREMSO in our cohort of MS patients, suggesting that increased CSF GFAP concentrations may be associated with a higher risk of conversion to secondary progressive MS.

It has been reported that immunosenescence is characterized by changes in both the adaptive and innate immunity [62]. Aging is characterized by an excess of inflammatory molecules which promotes chronic low-grade systemic and CNS inflammation. This state, also defined as “inflammaging”, may favour proinflammatory changes in resident immune cells as microglia and astrocytes, leading to chronic compartmentalized CNS inflammation that characterizes progressive MS phenotypes.

To test the hypothesis that higher CSF inflammation, microglial, and astroglia activation may predispose to inflammatory neurodegeneration, we explored possible correlations between the CSF levels of GFAP and sTREM2, and structural MRI measures. A negative correlation was found between GFAP and cerebellar volume at the time of diagnosis, conversely no significant associations emerged with white matter lesion load. Previous studies reported that serum GFAP levels correlated with lesion load and were negatively associated with WM and GM volumes [34] and that GFAP levels correlated with infratentorial lesion load [12]. Our results agree with an association between astrocyte activation and increased neurodegeneration. Notably, astroglial activation by proinflammatory molecules has been involved in synaptic hyperexcitability and excitotoxic damage in the cerebellum of EAE mice [63]. The role played by astrocytes in inflammatory-driven neurodegeneration may therefore explain their involvement in MS progression.

The main limitations of the present study are the small number of participants, particularly for MRI analysis, and the lack of a control group. However, this is the first study evaluating the correlation between GFAP, sTREM2, and cytokine CSF levels in the early stages of RRMS. Further studies are needed to unveil specific associations between markers of microglial and astroglial activation and cytokines subsets, also exploring other important astrocytic (i.e., YKL-40, S100) and microglial (i.e., CD11b, CD45, Iba1) biomarkers. In addition, prospective studies are needed to assess the impact of the CSF levels of GFAP and sTREM-2 on the MS disease course, and particularly on the risk of secondary progression.

## 5. Conclusions

In conclusion, the correlation between GFAP, sTREM-2, and CSF cytokines, may indicate an association between central inflammation and microglial and astroglial pathological reactivity, in the early MS stages. Our results suggest that GFAP and sTREM-2 represent suitable biomarkers of central inflammation in MS. Increased CSF expression of GFAP at the time MS diagnosis may characterize patients with a higher risk of progression.

## Figures and Tables

**Figure 1 biomolecules-12-00222-f001:**
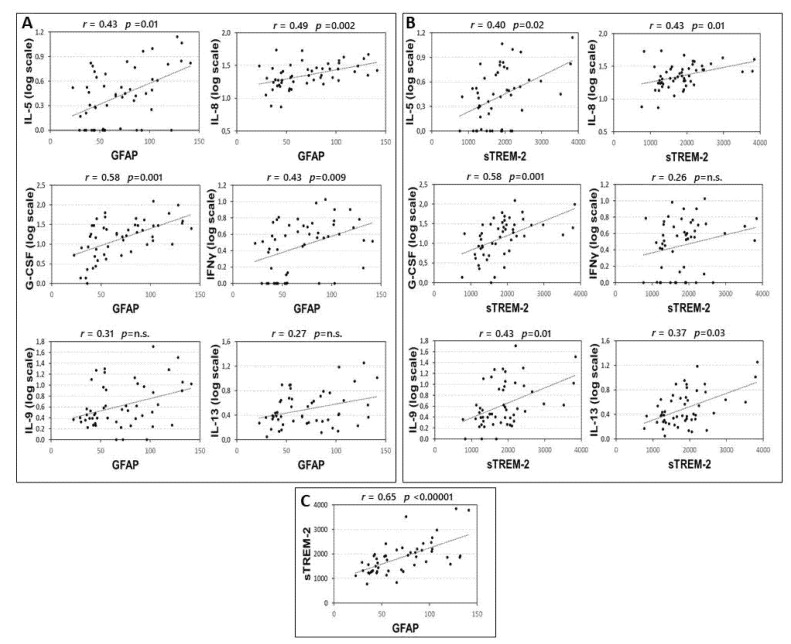
Correlation between CSF levels of GFAP, TREM-2 and inflammatory cytokines. Spearman’s correlations between CSF levels of GFAP and cytokines (**A**), sTREM2 and cytokines (**B**) and GFAP and sTREM2 (**C**). *p*-value is shown according to Benjamini-Hochberg (B-H) correction. Median concentration (min-max), pg/mL, of IL-5: 1,74(0–12,85); IL-8: 21.23(6.38–53.60); GCS-F: 15.59 (0–122.70); INF-γ: 2.45 (0–9.66); IL-9: 2.20 (0–50.33), GFAP: 65.78 (22.72–141.44); sTREM-2: 1858.25 (767.08–3839.92). Abbreviations: Cerebrospinal fluid (CSF); glial fibrillary acidic protein (GFAP); soluble triggering receptor expressed on myeloid cells-2 (sTREM2); interleukin (IL); granulocyte colony stimulating factor (G-CSF); interferon gamma (INFγ); Spearman’s rho (r); B-H *p*-value (*p*).

**Figure 2 biomolecules-12-00222-f002:**
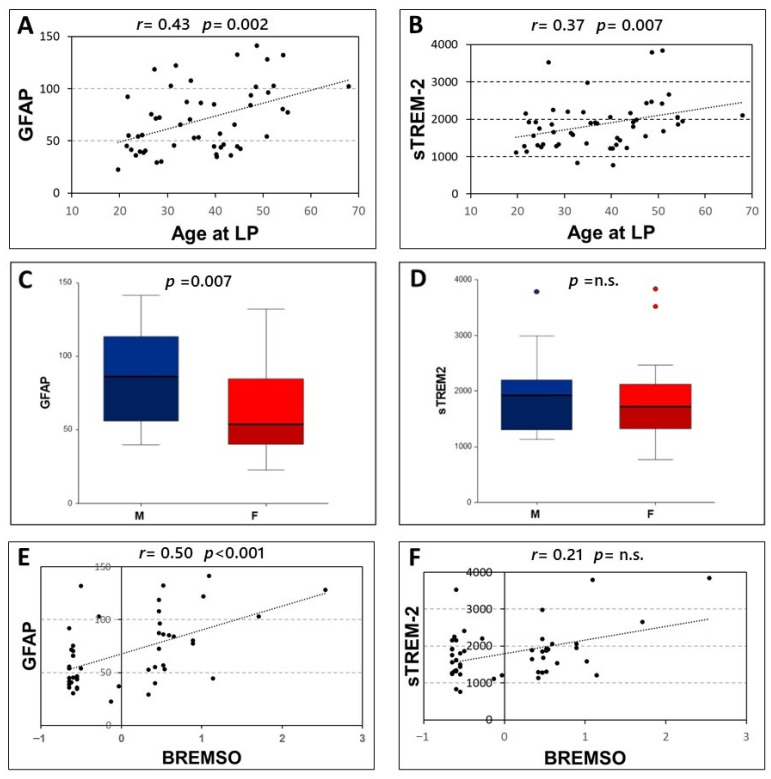
Correlation of CSF levels of GFAP and sTREM2 with clinical characteristics. Spearman’s correlations between CSF GFAP and sTREM2 and age at lumbar puncture (**A**,**B**) and BREMSO (**E**,**F**). Comparison between CSF GFAP or sTREM2 levels in male and female patients (**C**,**D**), Mann–Whitney U-test. Abbreviations: Cerebrospinal fluid (CSF); glial fibrillary acidic protein (GFAP); soluble triggering receptor expressed on myeloid cells-2 (sTREM2); Lumbar puncture (LP); male (M); female (F); Bayesian Risk Estimate for Multiple Sclerosis at Onset (BREMSO); Spearman’s rho (r); *p*-value (*p*)

**Figure 3 biomolecules-12-00222-f003:**
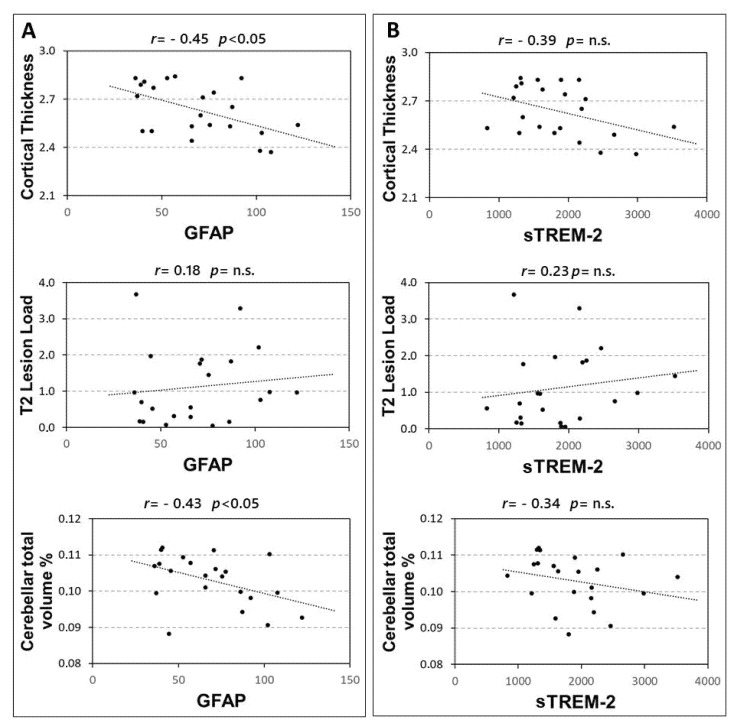
Correlations between CSF GFAP and sTREM2 and MRI structural measures. Spearman’s correlations between CSF GFAP and MRI structural measures (**A**) and sTREM2 and MRI measures (**B**). Abbreviations: Cerebrospinal fluid (CSF); glial fibrillary acidic protein (GFAP); soluble triggering receptor expressed on myeloid cells-2 (sTREM2); Spearman’s rho (r); *p*-value (*p*).

**Table 1 biomolecules-12-00222-t001:** Clinical characteristics of MS patients.

Variable	RRMS Patients (n = 51)
Sex, F	34 (66.7)
Age at LP, years	36.52 (27.31–45.28)
Disease duration, months	4.97 (1.72–28.97)
EDSS	1.5 (1–2)
OCB presence	41 (80.4)
BREMS at onset	−0.13 (−0.6–0.52)
Radiological disease activity	21 (41.2)
GFAP, pg/mL	65.77 (43.60–93.59)
TREM-2, pg/mL	1858.24 (1322.43–2152.68)
Cortical thickness, mm	2.62 (2.5–2.79)
T2 Lesion Load, mm^3^	0.86 (0.26–1.83)
Cerebellar Volume %	0.105 (0.099–0.108)

Data are given as number (%) for dichotomic variables, and as median (interquartile ranges) for continuous variables. Abbreviations: RRMS: relapsing remitting multiple sclerosis; F: female; LP: lumbar puncture; EDSS: expanded disability status scale, OCB: oligoclonal bands, BREMS: Bayesian Risk Estimate for Multiple Sclerosis, CSF: cerebrospinal fluid; GFAP: glial fibrillary acidic protein, sTREM2: soluble triggering receptor expressed on myeloid cells-2.

## Data Availability

Anonymized database are available upon reasonable request. Please contact the corresponding author.

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
