# Peer review of "Neuroinflammation Is Associated with GFAP and sTREM2 Levels in Multiple Sclerosis"

_biomolecules, 2022, doi:10.3390/biom12020222_

Round 1
Reviewer 1 Report
This manuscript describes a study investigating the association of GFAP and sTREM2 levels with inflammation-related molecules, clinical characteristics, and MRI measures in the CSF of MS patients who were newly diagnosed and treatment-naïve. The authors conclude that GFAP and sTREM2 are biomarkers of central inflammation in MS patients. In addition, enhanced CSF GFAP expression at the time of diagnosis may suggest a higher risk of progression. The study is interesting. My concerns are:
- To determine the importance of measuring CSF levels of GFAP and sTREM2, the authors need to determine whether these parameters correlate with MS progression and how treatments affect them.
- The “Introduction” does not provide sufficient inflammation of the previous studies on the relationship between CSF molecules and MS activities. It does not point out the new message that this particular study aims to add in this field.
- Line 143-160: The descriptions match the data in Table S1 but do not match Figure 1.
- Line 173-178: Figure 2C and D do not mean association. They only mean that male patients have higher levels of GFAP than female patients. In addition, the descriptions do not match the data in Figures 2A and B.
- Line 184-186: The descriptions do not match Figure 2.
Author Response
Q1. To determine the importance of measuring CSF levels of GFAP and sTREM2, the authors need to determine whether these parameters correlate with MS progression and how treatments affect them.
A1: Thank you. According to the Reviewer’s suggestion, we improved in the Introduction the description of previous literature regarding the association between GFAP/TREM-2 and disease course of MS (Page 2, lines 62-68). Furthermore, we have better specified that we included only MS patients at the time of diagnosis (page 2, lines 85-87) and that all patients enrolled were not treated with corticosteroids or DMTs before the clinical assessment and CSF collection (page 2, lines 96-97). In addition, no prospective clinical measures were collected in our study and further research is needed to assess the impact of these biomarkers on MS course (Page 10, Lines 360-362).
Q2. The “Introduction” does not provide sufficient inflammation of the previous studies on the relationship between CSF molecules and MS activities. It does not point out the new message that this particular study aims to add in this field.
A2: According to Reviewer’s comment, in the Introduction we have better described previous data on the correlations/associations between CSF inflammatory molecules and MS characteristics including disease activity. We have also specified that the association between CSF inflammatory molecules and microglial/astroglial biomarkers has not been previously investigated in MS (Page 2, lines 69-73).
Q3. Line 143-160: The descriptions match the data in Table S1 but do not match Figure 1.
A3: Thank you. We have corrected Figure 1. Following the comments of Reviewer 2 we have also removed from the text (paragraph 3.1) the results of Spearman’s correlations.
Q.4 Line 173-178: Figure 2C and D do not mean association. They only mean that male patients have higher levels of GFAP than female patients. In addition, the descriptions do not match the data in Figures 2A and B.
A4: Thank you for your comments. Concerning Figure 2C and D, in the caption we have specified that figure C and D are confrontational Mann Whitney U test, while the term “correlation” is referred to figures 2A,B,E,F. We also clarified the description in the text (page 5, lines 198-201)
Q5 Line 184-186: The descriptions do not match Figure 2.
A5: Thank you. We have now corrected the description of Figure 2
Reviewer 2 Report
I really liked the manuscript and it offers useful information on the association of GFAP and sTREM-2 with inflammatory markers in multiple sclerosis.
I have only two remarks:
1) It would be good to see the concentration range of the inflammatory cytokines. Interestingly, in a previous publication for example IL-5 was below detection limit, and they also used Bio-Plex kit (Immune Soluble Factors in the Cerebrospinal Fluid of Progressive Multiple Sclerosis Patients Segregate Into Two Groups, doi.org/10.3389/fimmu.2021.633167).
2) The correlations on pages 4-5 are difficult to follow in the text. Most of them have been presented in a table format in the supplementary information (which is a good idea). In my opinion, if the Spearman's correlations are presented in the table it is not necessary to repeat them in the text.
Author Response
Q1: It would be good to see the concentration range of the inflammatory cytokines. Interestingly, in a previous publication for example IL-5 was below detection limit, and they also used Bio-Plex kit (Immune Soluble Factors in the Cerebrospinal Fluid of Progressive Multiple Sclerosis Patients Segregate Into Two Groups, doi.org/10.3389/fimmu.2021.633167).
A1: Thank you. According to the Reviewer’s suggestions, we have added the concentration range (median and min-max values) of relevant cytokines in figure 1 legend.
Q2: The correlations on pages 4-5 are difficult to follow in the text. Most of them have been presented in a table format in the supplementary information (which is a good idea). In my opinion, if the Spearman's correlations are presented in the table it is not necessary to repeat them in the text.
A2: Thank you. Following the Reviewer’s suggestion, we have removed the textual description of results
Reviewer 3 Report
In the present paper by Azzolini et al. entitled “Neuroinflammation is associated with GFAP and sTREM2 levels in Multiple Sclerosis”, the reader is presented with a useful, first study evaluating the correlation between GFAP, sTREM2 and cytokine CSF levels in the early stages of RR-MS. Overall, the paper is written in a fluent way providing the data of CSF levels of GFAP and sTREM-2 and the concentrations of a large set of proinflammatory and anti-inflammatory molecules in a group of RRMS patients. However, there are a few concerns and/or comments that would require attention and I encourage the authors to address them before the paper is accepted for publication.
- In introduction the authors point to the cross-talk between microglia and astrocytes and that altered interaction between astroglia and microglia may be involved in the pathogenesis of different neurodegenerative conditions. The interplay of astrocytes and microglia may be given in MS pathogenesis and the underlying mechanism of neurodegeneration in MS.
- The authors investigated the levels of GFAP and sTREM-2 along with cytokine levels only in a group of 51 RRMS patients. They claim that results suggest that GFAP and sTREM-2 represent suitable biomarkers of central inflammation in MS. Did they also used the CSF of the control patients (without MS present) to show the increase in the GFAP and sTREM-2 levels compared to control group? This data might be important for the conclusion of the paper.
- They used specific marker for reactive astrocytes, GFAP; and not that much specific marker for activated microglia, s-TREM2. Did they also check the correlation with the CSF levels of GFAP with other microglial markers such as CD11b, CD45, Iba1?
- Analysis of CSF proinflammatory and anti-inflammatory cytokines and chemokines was performed using a Bio-Plex multiplex cytokine assay. A description of the data processing/analysis would be very welcome.
- In Conclusion, the meaning of correlation of the CSF levels of GFAP and sTREM-2 with inflammatory cytokines IL-8,+ and IL-5 is missing.
Author Response
Q1: In introduction the authors point to the cross-talk between microglia and astrocytes and that altered interaction between astroglia and microglia may be involved in the pathogenesis of different neurodegenerative conditions. The interplay of astrocytes and microglia may be given in MS pathogenesis and the underlying mechanism of neurodegeneration in MS.
A1: Thank you. We have modified the introduction according to the suggestion of the Reviewers (page 2, lines 53-75)
Q2: The authors investigated the levels of GFAP and sTREM-2 along with cytokine levels only in a group of 51 RRMS patients. They claim that results suggest that GFAP and sTREM-2 represent suitable biomarkers of central inflammation in MS. Did they also used the CSF of the control patients (without MS present) to show the increase in the GFAP and sTREM-2 levels compared to control group? This data might be important for the conclusion of the paper.
A2: Thank you for this important comment. In the discussion (Page 10, line 355). we have now added a comment concerning the absence of a control group, which represents a limitation of the present work and modified the conclusions.
Q3: They used specific marker for reactive astrocytes, GFAP; and not that much specific marker for activated microglia, s-TREM2. Did they also check the correlation with the CSF levels of GFAP with other microglial markers such as CD11b, CD45, Iba1?
A3: We have now specified in the discussion that other important microglial biomarkers have not been investigated in the present paper (Page 10, lines 358-360), and further studies are needed to clarify possible associations with inflammatory biomarkers.
Q4: Analysis of CSF proinflammatory and anti-inflammatory cytokines and chemokines was performed using a Bio-Plex multiplex cytokine assay. A description of the data processing/analysis would be very welcome.
A4: Thank you. We have now better detailed the description in the methods section (page 3, lines 105-107)
Q5: In Conclusion, the meaning of correlation of the CSF levels of GFAP and sTREM-2 with inflammatory cytokines IL-8,+ and IL-5 is missing.
A5: Thank you. We have now tried to improve the discussion on the correlations between GFAP/TREM-2 and CSF cytokines. We have also modified the conclusions, specifying that “the correlation between GFAP and sTREM-2 CSF levels and inflammatory molecules, in early MS stages, may indicate an association between central inflammation and microglial and astroglial pathological reactivity” (page 10, lines 365-367)
Round 2
Reviewer 1 Report
- Line 161-162: IFN-gamma data (significant correlation) is not mentioned in the text.
- There are still several typos and grammatical errors.
Author Response
Thanks again for your comments and suggestions.
According to the Reviewer's remarks:
- the results section has been corrected (line 161)
- typos and grammatical errors have been revised in the text.